# The Friendly Health Issue Network to Support Computer-Assisted Education for Clinical Reasoning in Multimorbidity Patients

**Fabrizio Pecoraro** [1] , **Fabrizio L. Ricci** [1,2] , **Fabrizio Consorti** [3,4] , **Daniela Luzi** [1] **and Oscar Tamburis** [1,5,*]

1   Institute for Research on Population and Social Policies, National Research Council, 00185 Rome, Italy;
    f.pecoraro@irpps.cnr.it (F.P.); f.ricci@irpps.cnr.it (F.L.R.); d.luzi@irpps.cnr.it (D.L.)
2   e-Health Virtual Laboratory, National Research Council, 00185 Rome, Italy
3   Department of Surgical Sciences, Sapienza University of Rome, 00185 Rome, Italy;
    fabrizio.consorti@uniroma1.it
4   Italian Society of Medical Education, 00185 Rome, Italy
5   Department of Veterinary Medicine and Animal Productions, University of Naples Federico II,
    80137 Naples, Italy
*   Correspondence: oscar.tamburis@unina.it

**Abstract:** Clinical reasoning in multimorbidity conditions asks for the ability to anticipate the possible evolutions of the overall health state of a patient and to identify the interactions among the concurrent health issues and their treatments. The HIN (Health Issue Network) approach, as Petri Nets-based formal language, is introduced as capable of providing a novel perspective to facilitate the acquisition of such competencies, graphically representing the network among a set of health issues (HIs) that affect a person throughout their life, and describing how HIs evolve over time. The need to provide a more immediate user-oriented interface has led to the development of f-HIN (friendly HIN), a lighter version based on the same mathematical properties as HIN, from which stems in turn the f-HINe (friendly HIN extracted) model, used to represent networks related to either real patients' clinical experiences extracted from electronic health records, or from teacher-designed realistic clinical histories. Such models have also been designed to be embedded in a software learning environment that allows drawing a f-HIN diagram, checking for its format correctness, as well as designing clinical exercises for the learners, including their computer-assisted assessment. The present paper aims at introducing and discussing the f-HIN/f-HINe models and their educational use. It also introduces the main features of the software learning environment it was built upon, pointing out its importance to: (i) help medical teachers in designing and representing the context of a learning outcome; and (ii) handle the complex history of a multimorbidity patient, to be conveyed in Case-Based Learning (CBL) exercises.

**Keywords:** clinical reasoning in multimorbidity; medical education; Petri Nets; Health Issue Network; educational software; Case-Based Learning; soft system methodology; Living Lab

## 1. Introduction

A vast literature is available on the theories [1,2] and educational methods [3,4] for clinical reasoning. Most of these models and methods focus on an acute condition or the onset of a chronic health issue. Nevertheless, the epidemiology in most countries has changed, with an increasing prevalence of chronic conditions and—moreover—multimorbidity [5]. Conceptual and organizational models for chronic care and multimorbidity are available [6,7], but there is a lack of pedagogies for the development of the ability of clinical reasoning in chronic multimorbid conditions. For this kind of patient, a shift is requested from the traditional disease-oriented approach to a more functional patient-tailored one [8,9]. In our approach, the construct of clinical reasoning in multimorbidity (CRM) encom-

passes the ability to understand the evolution over time of the health issues, their mutual interaction, and of anticipating the evolution of the whole state of a patient.

A broader approach in terms of medical education, medical research, and special healthcare delivery is needed for patients with multimorbidity [10,11]. Many efforts have been made to generate guidelines for the most common possible multimorbidity conditions, and to report and broadcast the available evidence-based medical knowledge. In the last decades, part of the medical informatics community has, therefore, been working extensively to develop computing structures that could be exploited to convey such knowledge, to support the educational needs of clinicians and medical students [12–14]. In particular, Petri Nets have widely been used in the medical domain for workflows modelling [15,16], access to services management [17], resource management [18], or guidelines [19], to figure out the flow of planned and executed activities to accomplish specific tasks, or to solve clinical problems. On the other hand, the deployment of technology-enhanced education has been in evolutionary development since the second half of the last century, and it is currently acknowledged as playing a pivotal role in delivering education to health professionals for the development of knowledge, skills, and behavior [20,21]. The present paper: (i) briefly reviews the theoretical background of computer-assisted education of clinical reasoning, and shows that CRM is a new concept, currently unsupported by any technology-enhanced pedagogy; (ii) describes the main features of the PN-based Health Issue Network (HIN) approach [22], whose underlying mathematical properties ensure a sound and formally correct representation of the evolutions over time of the patient's Health Issues (HIs); (iii) introduces the f-HIN model (along with its derivation, called f-HINe), as a set of user-friendly methods and tools to support undergraduate medical education for CRM, via the design and development of exercises aimed at reconstructing the network of diseases, with specific reference to multimorbidity patients; and (iv) points out in its principal functions a tailor-made software learning environment for the design, validation, and evaluation of f-HINe networks for educational purposes, called fHINscene.

The paper is organized as follows: after the Introduction, Section 2 provides a theoretical background for what concerns the concepts of clinical reasoning, and the application of computer-based education for health sciences; Section 3 introduces HIN as a Petri Nets-based educational tool to represent the patient's clinical history; Section 4 provides a brief description of f-HINe model features; Section 5 describes the fHINscene software focusing on the editor and comparator modules; discussion and conclusions are eventually provided, respectively, in Sections 6 and 7.

## 2. Theoretical Background

### 2.1. Clinical Reasoning

Clinical reasoning is the overall cognitive process through which a diagnosis is made from the available information (symptoms, signs, diagnostic test results). Clinical reasoning drives also the choice of treatment and the overall process of management of a chronic condition. The dual-process theory is currently considered the model that best describes diagnostic clinical reasoning. It takes place through two processes, distinct but in a circular relationship: hypothetical-deductive reasoning and intuition, the latter based on pattern recognition [23–25]. The intuitive recognition is based on mental constructs in the form of typical frameworks (named illness scripts), which are compact lists of characteristics [26]. A diagnosis is the act of classifying the patient's condition within a nosology class. Although the diagnostic reasoning process is aimed at defining any kind of condition, the examples in highly cited articles on clinical reasoning are constantly about acute or emergency conditions [27–29] and the dual-process theory implicitly considers only the diagnosis of a single disease. Nevertheless, as anticipated in the introduction, in most countries, the epidemiology has changed toward a high prevalence of chronic and multimorbidity conditions. The problem is often not to do a diagnosis, but to manage a situation in which a set of concurrent chronic diseases evolve and interact. Time seems to be the great absentee in the current models of clinical reasoning. In a review of literature

about clinical reasoning in multimorbidity, driven by the principles of the PRISMA Extension for Scoping Reviews [30], and preliminary to the development of the HIN model, only three articles explicitly dealt with clinical reasoning in multimorbidity. Cook et al. [31] addressed the concept of management reasoning which "involves negotiation of a plan and ongoing monitoring/adjustment of that plan". Audétat et al. [32] have just proposed a study protocol to inquire the reasoning process in multimorbidity. Their starting point is that "the main goal of these clinical reasoning processes is the search for a balance between the evidence-based care options, the patient's priorities and trying to withhold their quality of life, through the consideration and prioritization of several possibilities". Finally, Jayasinghe [33] proposed a method called Clinical Reasoning Map, in which a diagram network of nodes and arrows is used to represent the mutual relationships among different diagnoses. In these three examples, the action is always intended in the present moment, and the concept of evolution over time of the health issues is missing or at least not explicitly represented.

### 2.2. Computer-Based Medical Education

The original concept of computer-assisted instruction has also been referred to as computer-based learning, computer-based education, or computer-assisted, among others. Nevertheless, due to the continuous evolution of concepts, pedagogies, and theories, as well as to the growth and spreading of methods, features, technologies, and delivery settings, a clear denomination and categorization in the literature for this kind of health education is still lacking [34]. To this regard, it is worth noticing that in many cases, this kind of education has only been considered a way to deliver specific contents via a number of supports (e.g., computer discs/CD–ROMs, digital versatile discs/DVD–ROMs, external hard discs, universal serial bus/USB memory sticks, different software packages, or the Internet) for the students to be able to follow self-paced directed learning, thus, progressively supplementing or even replacing the classic face-to-face learning programs [35]. According to [36], computer-based education is comprised within the wider technology-enhanced learning (TEL) category, which concerns the use of technologies to support learning, whether the learner is local (i.e., on-campus) or remote (at home or in the workplace). Strong evidence is available about the effectiveness of TEL [37] and computer-based education, especially in the field of clinical reasoning, through simulation programs often indicated as virtual patients [38]. Virtual patients are a tool framed in the overall approach called Case-Based Learning (CBL), in which the learner is challenged with a real or realistic clinical case and asked to interpret, decide, and manage it. CBL has proved to be highly effective in transferring theory to practice [39]. In this sense, TEL can also be considered as a specific form of digital health education, introduced by [40] as the delivery of educational material through Information and Communication Technology using a wide variety of pedagogical designs and formats for clinical purposes.

Side effects of digital learning have also been registered: as the current CoVid-19 pandemic situation has caused a massive recourse to electronic lectures and programs for medical studies as well, many students have been affected by mental health and burnout issues [41]. This points out, once again, the critical aspects needed to achieve a viable integration between so–called "modern" and "classic" educational approaches for a thorough training of healthcare professionals.

## 3. HIN: A Formal Background

### 3.1. The Petri Nets for Modelling Health Evolutions

A Health Issue (HI) of an individual, also denotable as a clinical condition, can be referred to as a disease hypothesis, a sign/symptom, a diagnosis, a risk factor, or any other piece of clinical information. A HI network (HIN) describes the health status of an individual throughout their life, thus, capable to highlight how, e.g., (i) each condition has changed over time; (ii) the interactions between different conditions have influenced their

evolutions; (iii) a given treatment plan for a specific condition may have turned over time into a structured treatment pathway.

In the HIN model, a clinical condition can: (i) evolve (spontaneously or after treatment) either to worsen or to improve; (ii) generate another clinical condition (although remaining active) as a complication/cause or catalyse as co-morbidity the evolution of another problem; (iii) relapse after resolution. It is also possible for a clinical condition to carry out an examining in-depth evolution, which points out the shift from: (i) the symptom reported by the patient, or a sign detected by the physician, to a diagnostic hypothesis or a diagnosis; (ii) a diagnostic hypothesis to a diagnosis, using a diagnostic test (laboratory, imaging, functional); (iii) a diagnosis to another one, whereas the first one turned out to be incorrect. From a teaching viewpoint, the clinical history of a patient, to be modelled via HIN, has the following characteristics [42]:

1.  It is based on two fundamental concepts, HI and evolution. The transition from one HI to another occurs through a well-defined evolution;
2.  It is generated by the entire additional set of evolutions connecting the patient's HIs. At any given time, the set of active HIs identify the specific health status of the patient: this implies that the clinical history can be considered as a linear system;
3.  It can comprise independent sub-histories, therefore, it can be considered as a discrete distributed system: the evolution of an HI may be independent of (or may overlap with) another evolution;
4.  It is an asynchronous system because even though concurrent evolutions can occur, HIs are only accounted to evolve one per time. Accordingly, every single evolution is only capable to partially and locally affect the whole clinical history;
5.  It is a system without memory because the identification of the new potential evolutions from a given health status does not depend on how such status has been reached.

Given these premises, an effective alignment was acknowledged between the requirements of the HIN model and the properties of a Petri Net (PN). As widely introduced by [43,44], a Petri Net is a direct graph with two types of nodes, places and transitions, connected by directed edges. An edge can only join nodes of different types, so there can be edges between places and transitions, but not between two places, or between two transitions. The place node that ends in a transition is called transition input place; a place node to which an edge arrives from a transition is called transition output place. Place nodes can contain a variable number of tokens. The distribution of tokens over the set of places in the PN network is called marking. Transitions act on incoming tokens according to a so-called firing rule. A transition is enabled, if it can be triggered, i.e., if there are tokens in all input places. When a transition is triggered, it consumes tokens from its input place and places a specified number of tokens in each of its output places. The HIN formalism was based on PNs' one, more precisely expanding the features of the predicate/transition (P/T) model (a generalization of the place/transition concept) [43,44], with some clinical condition evolution-related constraints. The main advantages offered by PNs to other types of state machines are: (i) easy modifiability (adding elements, places/transitions) without the need to start over and without an "explosion" of complexity; and (ii) modularity whereby the graph is expandable by assembling sub-models (see, e.g., [45]).

### 3.2. Main Features of HIN

The HIN model is characterised by two main concepts: (i) PNs' place nodes refer to the health issues (HIs) [22]; and (ii) PNs' transition nodes describe the previously identified evolutions between HIs. Detailed information describing both places (HIs) and transitions (evolutions) are collected in specific data forms which state: (i) for each HI, the clinical data characterising the problem and the diagnostic–therapeutic procedures initiated due to the presence of the HI itself; and (ii) for each evolution, the threshold values of clinical parameters, which identify the evolution itself. Another important feature inherited by the Petri Net paradigm is the possibility of marking each HI node with a token: this means that the patient is interested in that specific issue at a precise moment. The distribution

of tokens over the HIs work out a configuration of the network—called marking—that represents the current overall health status of the subject. Similar to what happens for PNs, a transition/evolution is considered enabled and can be fired when all its input HIs contain a token. Tokens migrate from all the input nodes of the transition to all its output nodes, thus, setting up a new configuration of the network that represents the new current health status of the subject. Since each HI node can contain only one token per time, HIN is technically a 1-bounded Petri Net or safe net. Moreover, the execution of the HIN is non-deterministic: when multiple transitions are enabled at the same time, they will fire in any order.

The adoption of PNs' formalism provides HIN with the capability to evaluate anonymized real/realistic cases and to generate timely exercises that make possible an effective discussion on the clinical case itself. For instance, the reachability graph can be obtained from the network to work out the set of health issues that can be reached from a specific patient's health state. Thanks to the properties of PNs, it becomes possible to make questions concerning: (i) reachability, i.e., which HIs are reached starting from a certain state (set of HIs); (ii) coverage, i.e., which HIs were necessary to reach a certain HI (or: which HIs the patient's history evolved through to reach a certain HI); and (iii) liveness, i.e., what connection exists between two HIs. The deployment of the HIN model allows us as well to measure the "distance" between two diagrams. This problem can be solved by rewriting a PN (e.g., by eliminating a place/HI node or a transition/evolution). For example, given a set of HIs (without the corresponding evolutions) indicating the initial and final health status of a subject, the learner is asked to indicate the possible evolutions of the health status during the time, i.e., to draw the evolutions and influences and link them to the corresponding HIs. The distance to be measured is, therefore, between the HIN diagram drawn by the learner, and the one originally figured out by the medical teacher as the correct solution to the exercise.

For a more detailed discussion of the HIN model, see [46].

## 4. The f-HINe Model

The risen awareness about the possible difficulties for medical teachers and students to "dive" into the graphic formalism of HIN, has led to the development of a "friendly" version of HIN (or f-HIN) as a lighter version of the original model, although based on the same mathematical properties. This process was performed according to the specifics of the Living Lab approach [47], to which the creation of a first version of the software system (fHINscene) introduced in this work can be related as well.

The f-HIN model can both represent a real clinical history from a generic subject extracted from a medical record, and "possible" stories featuring potential alternative evolutions for some of the HIs involved. The f-HINe model (e stands for extracted) is instead the friendly version of the HIN model derived from f-HIN and deployed to represent only clinical stories of either real patients or from teacher-designed realistic clinical histories. In both cases, the network reproduces a story that already occurred (or is acknowledged as occurred), where there are no alternative evolutions but only well-defined paths. These cases can be extracted from the electronic health record (EHR) of, e.g., a general practitioner [48] to meet specific learning objectives. The cases are anonymised and can be enriched with side comments, references, etc., included in the mentioned data forms associated with the elements of the model. In addition, given the real/realistic nature of the cases analysed with the f-HINe model, the situations described reflect the actual dynamics of the healthcare professionals' daily work, which may also include incomplete clinical information, incorrect diagnostic hypotheses, etc.

The f-HINe model is a diagram composed of nodes (representing the HIs) and edges (representing the evolutions from input HIs to output HIs). Edges can be drawn via: (i) a solid line when the HI evolves without altering its nature (e.g., recurrence, worsening, improvement, examining in-depth); (ii) a dashed line whereas a HI generates a new HI (e.g., complication, cause). In case of more input and/or outputs HIs for a single common evolution, the use of a static branch node (or aggregator) can be used as a point where

more input HIs converge, and/or from which more output HIs depart: this is the case of, e.g., a worsening or a complication in presence of a co-morbidity. Evolutions are, therefore, always labelled, and their related descriptive sheets report information about the activities performed during the diagnostic–therapeutic process. The whole set of activities associated with the evolutions describes the actual treatment process the patient has undergone. Table 1 shows the graphic primitives of the f-HINe model according to the fHINscene software (that will be introduced in the next section), along with specific clinical examples.

**Table 1.** Graphic representation of the f-HINe primitives.

| Type of Evolution | f-HINe Primitive | Examples |
|---|---|---|
| Recurrence |  | A = acute diverticulitis<br>Evolution = recurrence<br>Meaning = the patient had recurrent episodes of acute diverticulitis |
| Worsening/Examining in-depth/Improvement |  | A = 2nd degree kidney failure<br>B = 3rd degree kidney failure<br>Evolution = worsening<br>Meaning = a 2nd degree kidney failure worsened into a 3rd degree<br>A = abdominal pain<br>B = acute appendicitis<br>Evolution = examining in-depth<br>Meaning = an abdominal pain was interpreted as an acute appendicitis |
| Complication/Cause |  | A = diabetes<br>B = diabetic foot<br>Evolution = complication<br>Meaning = diabetes complicates with a diabetic foot |
| Worsening with co-morbidity |  | A = mild dementia<br>B = severe dementia<br>C = pneumonia<br>Evolution = worsening with comorbidity<br>Meaning = dementia of a patient worsened after the onset of a pneumonia |
| Complication with co-morbidity |  | A = peripheral artery disease<br>B = gangrene of the foot<br>C = heart failure<br>Evolution = complication with comorbidity<br>Meaning = the peripheral artery disease of a patient complicates with gangrene of the foot after an episode of heart failure |

An f-HINe diagram has the following constraints:

- f-HINe can have isolated nodes (i.e., non-developing HIs);
- Minimum f-HINe consists of only one HI node;
- f-HINe is a diagram with direct edges;
- f-HINe can be unconnected, i.e., made up of both several connected diagram parts (one per each developing HI) and isolated nodes;
- f-HINe features no cycles, except for recurrences;
- The static branch node is only an intermediate node and is always connected to HI nodes;
- Between two HIs there can at most stand one and only one evolution.

To improve the readability of a f-HINe diagram, each HI can be assigned a specific colour, according to for example its relation to a well-defined anatomical apparatus/body system, as described in the ICPC2 classification developed for the continuity of care [49]: the aim is to highlight, for each clinical condition considered, the clinical viewpoint of its evolutionary pathway. With specific reference to Figure 1, red colour identifies issues that do not interfere with the remainder clinical evolutionary pathway; green is for endocrinological issues; light blue means ocular issues; yellow marks instead immunological pathophysiological conditions. The attribution of colour to a class of HIs is arbitrary, yet it must be consistent in every HIN.

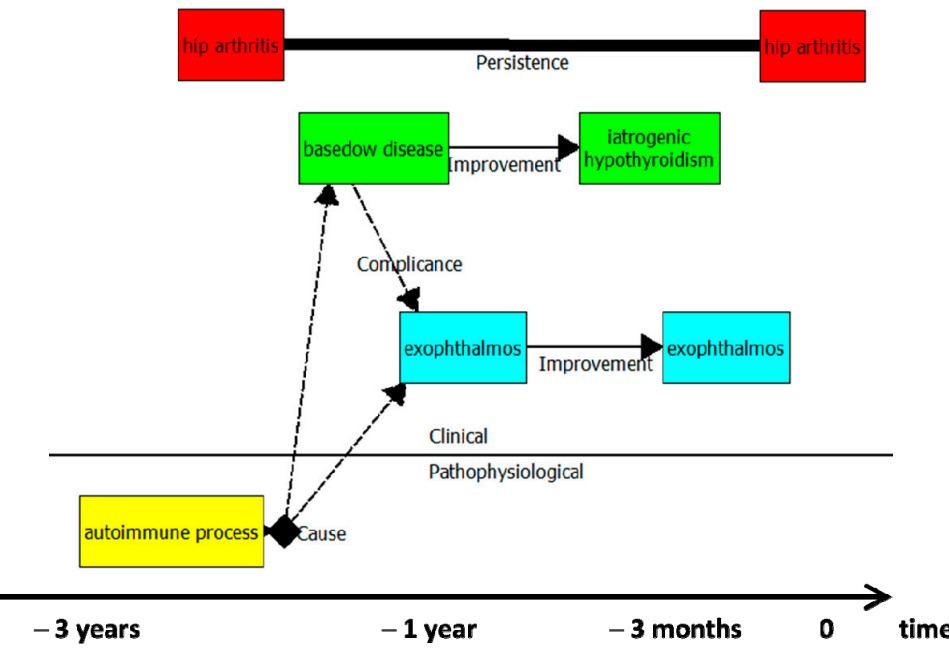

**Figure 1.** The primitives are linked to colours, levels, and time.

HIs in a f-HINe can be organised according to two different perspectives, each one represented within a level: clinical/semeiotic and pathophysiological. Each level provides a particular point of view on the clinical history: the overall synthesis (diagnoses as classes, risk factors as epidemiological knowledge); the data (symptoms, signs, other clinical information); the deep causal knowledge (ethology, pathophysiological correlations).

Furthermore, time plays an important role in the evolution of a clinical condition. In an f-HINe diagram, the problems are partially ordered: to this end, an implicit right-oriented time abscissa can be associated with the diagram. In case, e.g., of two HIs, the agreed rule points out the rightmost one as occurred later. The possible persistence of one or more conditions during time led eventually to the design of another primitive, depicted as a thick edge that connects the same duplicated HI.

Figure 1 shows the deployment of colours, levels, and time. It can be detected that:

- red colour identifies issues that do not interfere with the remainder clinical evolutionary pathway; green is for endocrinological issues; light blue means ocular issues; yellow marks instead immunological pathophysiological conditions;
- the autoimmune process (pathophysiological issue) stands as the cause of clinical conditions;
- the clinical history depicted span over three years.

Figure 2 shows the f-HINe diagram of Mario, a 76 yrs–old man who suffers from: (i) diverticular disease (green HIs); (ii) type 2 diabetes mellitus (yellow HIs); (iii) hip arthrosis (light blue HIs); (iv) benign prostatic hypertrophy (purple HIs); and (v) arterial hypertension (red HIs).

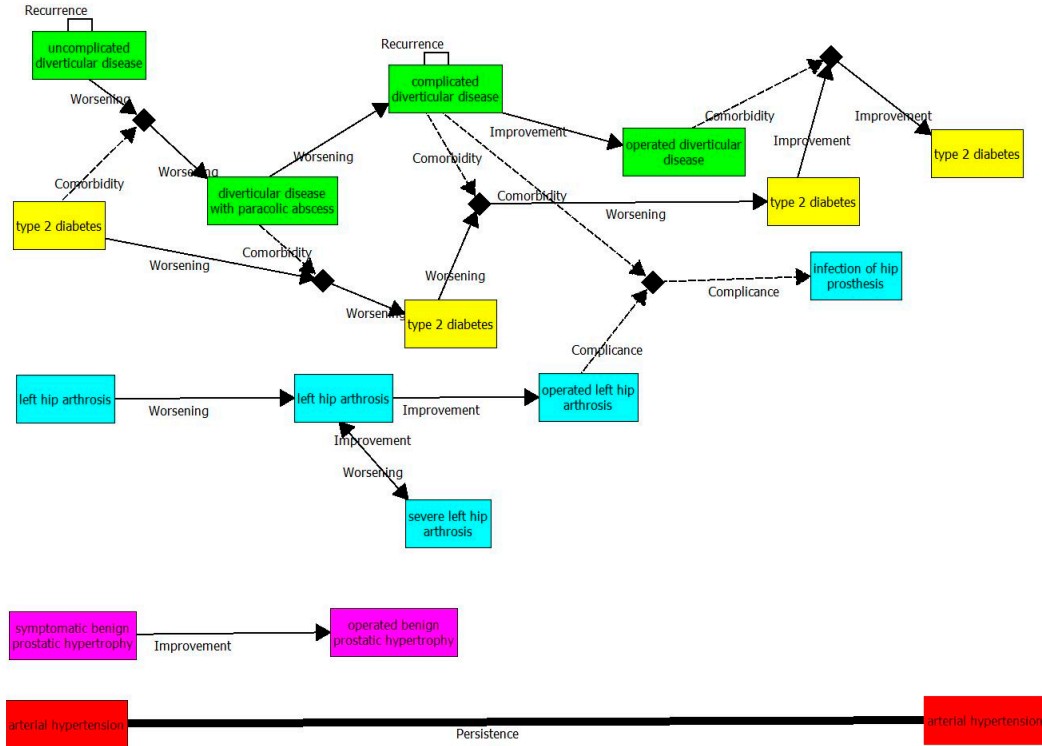

**Figure 2.** The f-HINe diagram of the patient "Mario".

Over the years, Mario's arterial hypertension, although pharmacologically controlled, has persisted; after surgery, the prostatic hypertrophy improved; hip arthrosis had episodes of improvement and worsening until surgery was considered, and the implantation of hip prosthesis led in a first moment to an improvement. In the same time interval, Mario, who had had past episodes of acute non-complicated diverticulitis, suffered a further acute episode with the development of a pericolic abscess. This came with a worsening of diabetes. The complicated diverticulitis then influenced the already worsened diabetes as well as the complication of the hip operation, which ended with the infection of the prosthesis. Two aspects worth highlighting are the following:

- Many details (the medication for arterial hypertension, the level of pain and muscle stiffness for the hip arthrosis) are "hidden" in the diagram but accessible to the user through the sheets associated with each HI and evolution;
- A difference exists between the evolution "recurrence" (see the non-complicated diverticular disease) and the "worsening-improvement" commuting dynamic (see hip arthrosis). While for the latter, the manifestations of the disease do not disappear, in the former each extemporaneous manifestation of the disease is independent of the previous/the following ones.

## 5. Software Application for The Design of f-HINe Diagrams

### 5.1. Introduction

This paragraph describes the main functionalities of the software application fHINscene developed to define, verify, and compare f-HINe diagrams. It has been developed using the QT cross-platform application framework [50] and based on the DiagramScene example of the Qt Toolkit. The class diagram reported in Figure 3 shows the main high-level classes highlighting two main functions: (1) editor (MainWindow) and (2) comparator (Evaluator).

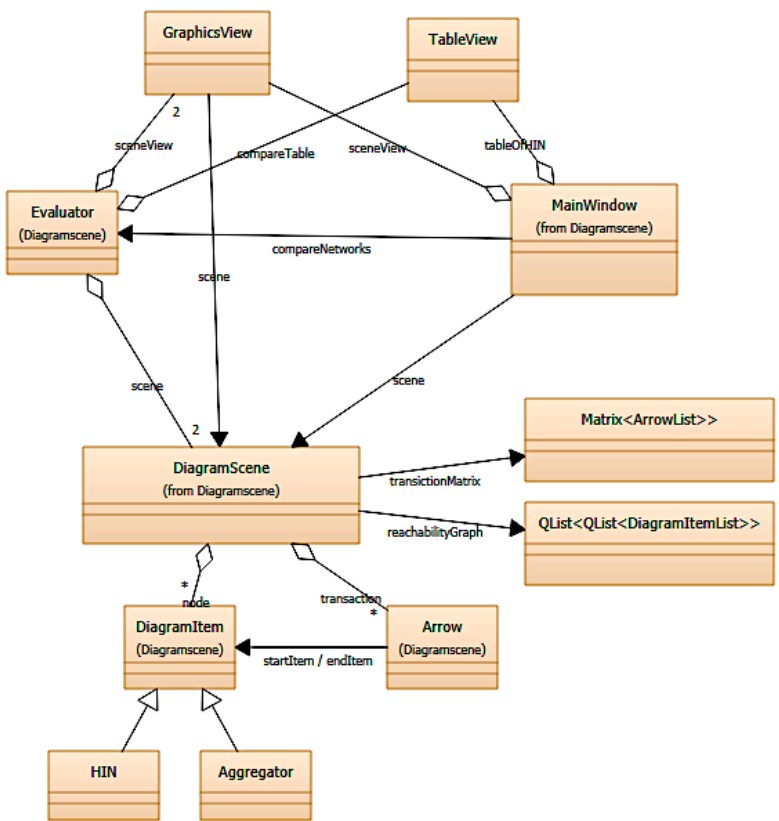

**Figure 3.** Class diagram highlighting the main application components (‡ highlights the multiplicity of the relevant association).

The editor supports teachers and learners in drawing a new f-HINe diagram as well as teachers in verifying its correctness, while the comparator support teachers in verifying the similarity of two f-HINe diagrams for educational purposes, simplifying the analysis of all f-HINe diagrams drawn by many students. Moreover, a third functionality foreseen in the project, but still under development, concerns a work environment for the development and execution of specific exercises based on the f-HINe model. Both editor and comparator have been implemented based on two graphic objects: GraphicsView, which represents the graphic component of the f-HINe model drawn and TableView, which instead serves to describe and summarize which the nodes that compose the diagram are and, in the case of the comparator, to highlight what the differences between the two f-HINe diagrams are. These graphic parts are closely related to modelling components, represented by the DiagramScene class adopted to define the f-HINe diagram. It is composed of a series of nodes (DiagramItem) and transitions (Arrow) that associate them. As highlighted in the previous paragraphs, each node can be either a health problem (HI) or a static branch node (Aggregator).

The graphics and management parts described above are completed by two main features: the transition matrix (Matrix) and the reachability graph (List of paths). In particular, the transition matrix, which is a slight modification of the Petri Nets' incidence matrix,

is composed of N rows and N columns (with N number of HI nodes). Each cell ($M_{i,j}$) contains the sequence of transitions that must be crossed to go from the health problem $HI_i$ to the health problem $HI_j$. Note that each cell can contain a list of paths as it does not take into account only the paths defined between two adjacent vertices but also those defined by a series of transitions (path) that connect the two HI nodes. The second feature is the reachability graph, which is one of the main foundations of Petri Nets. It collects the set of paths defined starting from the initial marking (the configuration of the network, i.e., source nodes) and the firing sequence necessary to reach the final configuration (i.e., well nodes). The transition matrix and the reachability graph are two essential features both for the comparison phase between two networks and to identify if the network respects the well-formed rules, the characteristics of a f-HINe diagram (see Section 3).

*5.2. Editor*

Figure 4 shows the graphical user interfaces (GUIs) of the implemented software, which displays the editor functionality.

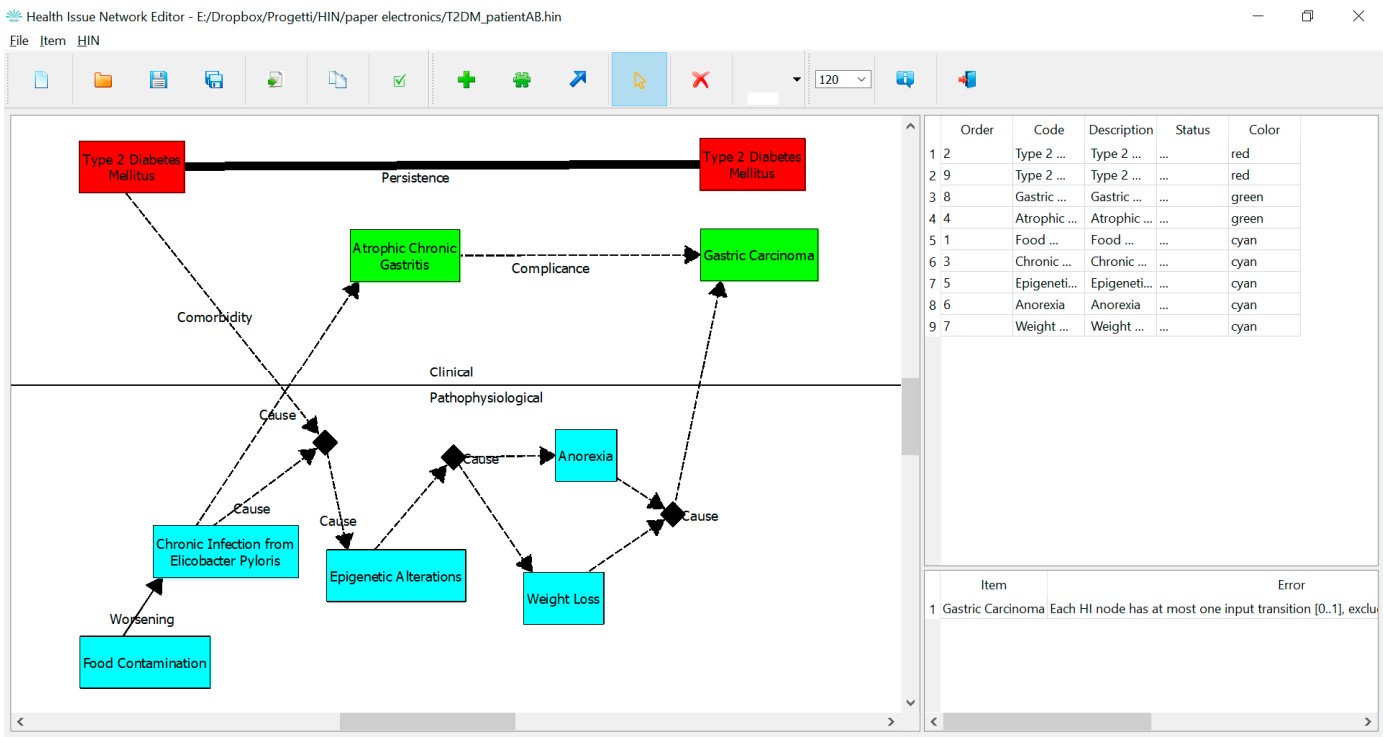

**Figure 4.** Screenshot of the main application window.

The main GUI (centralWidget) is divided into two main sections separated by a slidebar, one (on the left) represents the work environment where the user can draw the f-HINe diagram (sceneView), while the other, on the right, shows two tables both represented by the construct tableHIN. The upper one summarizes the different HI traced in the network with their characteristics: insertion order, HI code (e.g., ICD-9 coded [51] issue or free term), HI description (e.g., ICD-9 coded or free description), status (e.g., diagnostic hypothesis, etc.). Moreover, the colour used to represent each HI node is reported as it can be used to group pathologies. Finally, the status of the HI is reported as a stereotype in the HI box. The left side of the central widget shows the f-HINe representing the clinical story of the patient BC affected by T2DM, who has a chronic infection by Helicobacter Pylori. This condition evolves into an atrophic chronic gastritis. The pro-inflammatory environment due to diabetes makes epigenetic alteration more likely to happen. In the level of signs and symptoms, anorexia and weight loss occur. These symptoms, together with the known diagnosis of atrophic chronic gastritis, lead to the hypothesis/diagnosis

of gastric cancer. In a f-HIN diagram, the evolution "cause" has a wide semantic domain, which encompasses both the true causal relationship and the correlation (is associated to) or the phenomenological appearance (is manifested by).

The main functionalities of the application can be activated through the drop-down menus (i.e., File, Item, HIN, Element) and/or the relevant toolbar icons. In addition to the basic functions (e.g., opening a saved network, creating a new f-HINe diagram, saving the f-HINe diagram, exporting the f-HINe diagram to image) and the MACD functionalities (move, add, change, delete) of nodes and transitions, the software implements the check function of the network to capture if the network is well-formed according to the characteristics of a f-HINe diagram. As previously mentioned, this verification is done operating on the equivalent Petri net through the adoption of the reachability graph and the transition matrix. The network reported in Figure 4 contains only one error, as reported in the lower right table: gastric carcinoma has two input transitions: complication and cause. The reference diagram corrected by the error is shown in the left part of Figure 5.

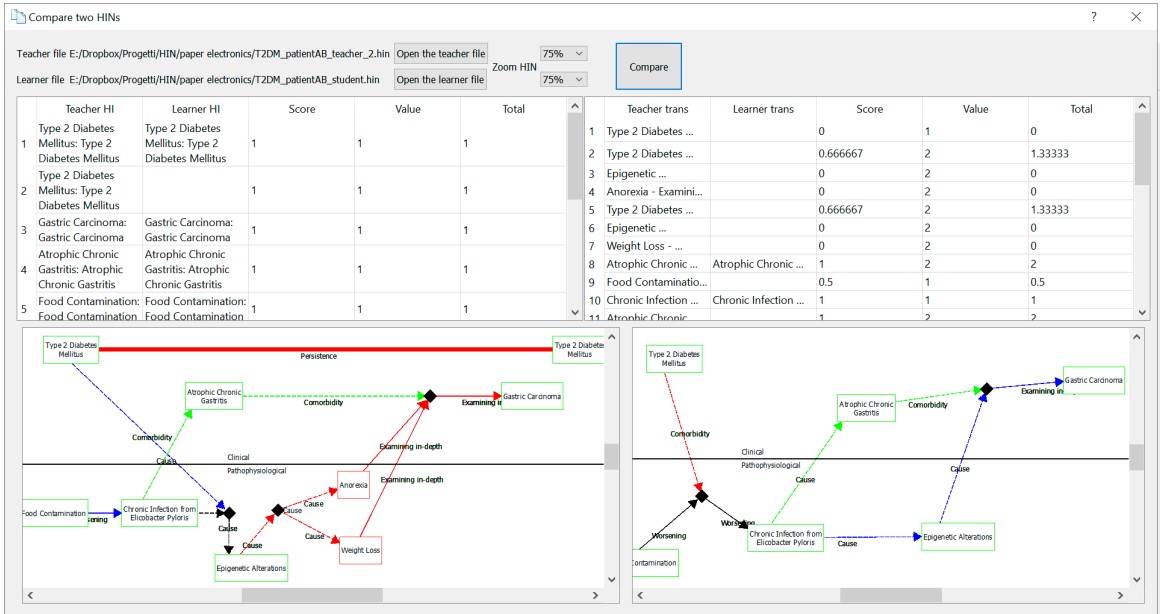

**Figure 5.** Screenshot of the application comparator window.

In its current release, the software application verifies the network adherence against the following rules:

- Each HI node has at most one input transition [0:1], excluding recurrences;
- Each HI node can have at most one integer output transition and unlimited dashed transitions (i.e., cause, complication, comorbidity);
- The aggregator node must have either two input or output transitions;
- The aggregator node must have at least one input and one output transition;
- If the inputs to an aggregator node are all dashed, then the output transitions are all dashed. If there is at least one solid input transition, then the output transitions are all solids;
- Input transitions of an aggregator must be of the same type (one type for dashed lines and one for solid lines);
- Output transitions of an aggregator must be either all solids or all dashed and report the same name;
- A transition cannot connect two aggregator nodes;
- A 'Persistence' transition can only connect two HI nodes;
- A 'Persistence' transition can only connect two HI nodes with the same health problem;
- Nodes connected to a 'Persistence' transition cannot have recurrences.

Moreover, even if the above-mentioned rules mainly check the static part of the network, two additional controls have been implemented considering its dynamic. Based on the reachability graph, the fHINscene verifies whether all nodes are traversed, and all transitions are fired. Table 2 shows the algorithm, written in pseudo-code, which determines what is the set of possible paths that go from the source HI nodes to the well HI nodes. This algorithm is based on the concept of recursion and uses two functions: computeReachabilityGraph, which represents the main function from which the recursive function fireTransition is then started. This fires the main transition that can be fired and recursively calling itself for defining the reachability graph. Note that, to define the reachability graph, following the Petri Nets philosophy, a transition can be fired if it meets the following two conditions: (i) all the source nodes contain the token; (ii) at least one destination node it is not marked. Note that, the algorithm verifies the firability of transitions based on their position on the network, starting from the left and moving to the right of the scene. For f-HINe, the firing rule remains the same as in the Petri Net (tokens must be present in all input HIs). The token moves from the input HI to the output H when it moves along the full edge: for the evolutions worsening, improvement, examining in-depth, recurrence. The token remains in the input HI and rises in the output HI in the case of a dashed edge to show the complication and cause evolutions.

**Table 2.** The algorithm in pseudo-code adopted to navigate the HIN and define the reachability graph.

| |
|---|
| **Function ComputeReachabilityGraph()** |
| 1.     insert the token in the source nodes of the graph, appending the set of source nodes to the current path |
| 2.     call the function **fireTransition(pathList)** * |
| **end function** |
| **function fireTransition(pathList)** |
| 1.     for each transition $t \in Tr$ |
|     1.1.     if $t$ is firable ** |
|         1.1.1.   fire the transition *** |
|         1.1.2.   update the token configuration |
|         1.1.3.   append the new configuration to the actual path |
|         1.1.4.   call the function **fireTransition(pathList)** |
|         1.1.5.   if the function **fireTransition** reports that no transitions have been fired |
|             1.1.5.1.create a copy of the actual path append it to the variable **pathList** |
|         1.1.6.   remove the last step from the last path |
|         1.1.7.   fire the transition in reverse **** |
|         1.1.8.   return to 1 and verify the subsequent transition |
| 2.     if all transitions have been checked and no one has been fired, return false else return true |

Legend: * pathList is a variable that contains all the paths and is iteratively updated during the navigation of the network. ** a transition is firable whether it matches the following rules: (1) it has not been fired yet; (2) all source nodes have a token; (3) at least one destination node is without the token. *** following the Petri Nets rules, fire a transition means to remove the token from the source nodes and insert a token in each destination node. The set of nodes with a token is called configuration. **** fire a transition in reverse means move the token from the destination nodes to the source nodes.

The same algorithm has been also adopted to define, for each HI node of the network, which sequence of transitions can be crossed, and which nodes can be reached. The application of this algorithm for all HI nodes of the network will be adopted to define the transaction matrix described in the previous paragraph. Note that, in this case, to determine if a sequence of transactions exists between two HI nodes and define the transaction matrix, the Petri Nets philosophy is slightly changed, and a transition can be fired if it meets the following two conditions: (i) the source HI node under investigation contains the token; (ii) the destination HI node is not marked. To perform this task, each transition $t_k$ is decomposed into a set of triples $<HI^s_i, HI^d_j, t_k>$ where $HI^s_i$ is the i-th source HI node,

and HI$^{d}_{j}$ is the j-th destination HI node. Starting from the f-HINe reported in the left part of Figure 5, the following triples can be detected:

{2} → {3}, worsening (Wo)
{3} → {4}, cause with comorbidity (Ca1)
{4} → {6}, cause (Ca2)
{4} → {7}, cause (Ca2)
{3} → {5}, cause (Ca3)
{1} → {4}, cause with comorbidity (Ca1)
{5} → {8}, examining in-depth with comorbidity (Ex)
{7} → {8}, examining in-depth with comorbidity (Ex)
{6} → {8}, examining in-depth with comorbidity (Ex)

The result of the execution of the above-described algorithm will produce the transition matrix reported in Table 3.

**Table 3.** Transition matrix defined on the basis of the f-HINe reported in the left part of Figure 5.

| | 1 | 2 | 3 | 4 | 5 | 6 | 7 | 8 |
|---|---|---|---|---|---|---|---|---|
| 1 | | | | {Ca1} | | {Ca1,Ca2} | {Ca1,Ca2} | {Ca1,Ca2,Ex} |
| 2 | | | {Wo} | {Wo,Ca1} | {Wo,Ca3} | {Wo,Ca1,Ca2} | {Wo,Ca1,Ca2} | {Wo,Ca1,Ca2,Ex}; {Wo,Ca3,Ex} |
| 3 | | | | {Ca1} | {Ca3} | {Ca1,Ca2} | {Ca1,Ca2} | {Ca1,Ca2,Ex}; {Ca3,Ex} |
| 4 | | | | | | {Ca2} | {Ca2} | {Ca2,Ex} |
| 5 | | | | | | | | {Ex} |
| 6 | | | | | | | | {Ex} |
| 7 | | | | | | | | {Ex} |
| 8 | | | | | | | | |

Legend: 1 = Type 2 Diabetes Mellitus; 2 = Food contamination; 3 = Chronic infection from helicobacter pylori; 4 = Epigenetic alterations; 5 = Atrophic chronic gastritis; 6 = Weight loss; 7 = Anorexia; 8 = Gastric carcinoma.

The initial marking is {Type 2 Diabetes Mellitus, Food contamination}. The final marking is {Type 2 Diabetes Mellitus; Chronic infection from Helicobacter pylori; Atrophic chronic gastritis; Gastric carcinoma}. The following possible evolution sequences (present in the reachability graph of the equivalent HIN graph) allow reaching the final marking from the initial one:

- Type 2 Diabetes Mellitus & Food contamination → Chronic infection from helicobacter pylori → Epigenetic alterations → Epigenetic alterations → Epigenetic alterations & Anorexia → Gastric carcinoma;
- Type 2 Diabetes Mellitus & Food contamination → Chronic infection from helicobacter pylori → Epigenetic alterations → Epigenetic alterations & Anorexia → Epigenetic alterations → Gastric carcinoma.

*5.3. Comparator*

Once the network has been created by a learner, it can be compared with a master/gold-standard network, for example, defined by the teacher. The comparison is based on the static part of the network by comparing two main concepts: HI nodes and transition. Considering HI nodes, the algorithm provides a one-to-one comparison to verify whether the HI nodes reported by the teacher have been also reported by the learner. Additional HI nodes can be considered depending on the description granularity of the storyline. Considering the comparison of transition both networks (i.e., teacher and learner ones) are broken down into a set of triples that describe the individual transitions of each treatment path. The presence of the triples in the teacher's network is then checked with those reported in the learner's HIN. To give a different weight to the single HI and the single transitions, the teacher can assign during the definition of the f-HINe diagram a value to each component of the network.

As shown in Figure 5, the comparison window of the application is composed of four distinct widgets: two scenes (i.e., bottom part of the window) showing the two f-HINe models (teacher on the left and learner on the right) and two tables (i.e., upper part of the window) showing the results of the comparison: on the left, the table reports the result of the HI analysis, while on the right the analysis of transitions is highlighted. To compute this comparison, the first step is to load the two f-HINs. This is done using the two buttons (i.e., open the learner file network, open the teacher file). Once the two HINs are loaded and displayed in the two scenes, the user can press the button *Compare*, which starts the comparison and checks what the differences between the two f-HINe diagrams are, both graphically and analytically.

From the graphic point of view, the perfectly matching nodes will be highlighted in green (i.e., Type 2 diabetes mellitus, Atrophic chronic gastritis, Gastric carcinoma, Chronic infection from helicobacter pylori, Food contamination, Epigenetic alterations), while those that are not present in the comparison network will be highlighted in red, such as Anorexia and Weight loss reported by the teacher and not by the learner. At the moment, the comparison is based on a Boolean function, which reports 1 only if the teacher HI and the learner's HI perfectly match. However, it is foreseen the introduction of suitable similarity algorithms to determine the distance between two HIs and showing a fuzzy similarity index between 0 and 1. This comparison could be based, for example, on the degree of kinship of two ICD-9 codes. However, this part has not yet been implemented since the concepts of similarity must be deepened especially considering that diagrams are not always defined using a nomenclator/vocabulary that supports this comparison. In a very similar way, the comparison between the transitions will be made and displayed. Moreover, in this case, perfectly matching transitions will be highlighted in green (e.g., Chronic infection from Helicobacter pylori that causes Atrophic chronic gastritis), while the objects not present in the comparison network will be shown in red, such as the persistency between the two Type 2 diabetes mellitus HIs reported in the teacher's network but not in the learner one. The presence of a path, even if not direct, is instead shown in blue such as the transition between the Epigenetic alterations and the Gastric carcinoma that is reported as a direct transition in the learner's network but present in the teacher's one only passing through Anorexia and Weight loss.

As previously mentioned, together with the graphic results, the software application provides two tables that summarize the result of the analysis in an analytical form comparing the HIs (see left table in Figure 5) and the transitions (see right table in Figure 5) of the teacher's and learner's networks. Both tables are composed of five columns reporting respectively: the object reported by the teacher as well as by the learner, whether the HIs match, the weighting value assigned by the teacher to the relevant object, the result of the comparison. The last rows of each table are devoted to objects reported by the learner and not present in the teacher's network, for this reason, both the value and the total values are not reported.

## 6. Discussion

This article describes the philosophy, methods, and tools of the f-HIN/f-HINe model, to represent the clinical history of multimorbidity patients. In this section, we compare our model to similar systems, highlighting the specificity of our model, and discuss the value of a collaborative approach and a soft systems methodology (SSM). Finally, we examine the educational value of our proposal.

The HIN approach provides students of medical sciences with an instrument that helps them learn how to deal with the whole complexity of a patient's clinical path, especially when chronic multimorbidity is present. The goal is to improve their CRM competencies, through exercises in which they represent in a well-structured formal way the clinical history of a patient with diverse concurrent diseases, and identify the overall set of HIs' interactions, as well as the types of relationships (evolutions) between the health issues. In this sense, HIN can be referred to as a non-hierarchical network with multiple inter-

connections, suitable to describe complex clinical scenarios (CCSs), to supplement lists of differential diagnoses (or problem lists), and to act as an aid in clinical management. Several system modelling approaches have been developed for healthcare, or have been borrowed from other science fields, which showcased specific capabilities to, e.g.,: describe specific aspects of a complex system, such as concept maps [52,53], or mind maps [54,55]; address instances from risk management, such as sequence diagrams [56–58], or; design care processes involving cross-organizational team working with multiple stakeholders along with complex transfers of information, medication, and patients [59]. None of these tools turned anyway capable of effectively addressing issues related to how activities may affect the change of a patient's health status, or how this evolves over time. Further, compared to them, the f-HIN/f-HINe model uses an extended set of types of evolution and explicitly represents concepts like time and evolution over time. The graphical appearance of the diagrams is intuitive for a medical student and powerful enough to let the user represent the same situation according to different points of view. The fHINscene warrants the drawing of formally correct diagrams, thus, allowing the exploitation of the computational properties of the underlying mathematical model of PNs. Premises similar to the HIN ones could only be retrieved in Jayasinghe's Clinical Reasoning Map [33] that deploys network diagrams with nodes used as clinical entities, and arrows as associations, to represent the interconnectedness of CCSs and to cater for a visual representation of systems science at the bedside. Nonetheless, this model is not based on a sound mathematical formalism, therefore, it does not guarantee either the correctness of the model, the navigation within it, or a suitable comparison between different models.

In their critical analysis, Jun et al. [60,61], although recognizing the importance of abstraction of real-world phenomena based on our information needs for effective quality improvement, found that many projects do not take into account the application of such models in the everyday line of work, especially because of the lack of prior/pre-work learning experiences of users, that would allow them to better perceive the intrinsic scopes of such methods. To this regard, the added value of f-HIN relates to the Living Lab approach it was built upon [62,63], which aims at figure out an educational path for medical sciences focused on: (i) a multidisciplinary, collaborative, user-centric perspective comprising both teachers and learners (end-user engagement); (ii) the seeking of actual fields of experimentation, such as CRM (real-life settings); (iii) the pursue of open innovation goals via continuous kaizen-based evolutionary thinking (progressive change) between the actors involved (user-driven innovation).

The complex adaptive nature of the healthcare sector requires the combined effort of experts from different domains to model healthcare processes for the overarching goal of quality improvement. In other terms, what Scally and Donaldson [64] named for the first time as "clinical governance" is just a piece of the much wider horizon outlined from the class of "system thinking" methods [65] that, via an appropriate approach to modelling healthcare, manage to address the mentioned complex nature of the sector [66,67]. Among these, the soft systems methodology (SSM) readily incorporates the complexities of the socio-technical nature of healthcare per se as well as with technology [68,69]. Petri Nets (PN) fall within the scopes of SSM—and accordingly is the f-HIN model, as PN–based formalism—because they allow analysts to account for several differing worldviews in an attempt to elicit as much information from a complex situation as possible [70]. In our case, the management of multimorbid conditions implies that different subjects may develop different personal lines of reasoning (worldviews) for what concerns the anticipation of the possible evolutions of the overall health state of a patient, as well as the identification of the interactions among the concurrent health issues and their treatments. This aspect emerges in f-HIN thanks to the CBL typology of exercises that involves the comparison between the network diagram designed by the doctor teacher (which creates it from a real/realistic case study) and those designed by the students. Such comparison is further enriched by the possibility to focus either on the clinical path of the patient (using the sole clinical and semeiotic levels) or on the actual evolution of the overall patient's health status (extending

also to the pathophysiological level). In the light of this, f-HIN turns, therefore, as a tool for reflective practice and abstract conceptualization that also encompasses Jayasinghe's systems science based on the "connectedness" of the human body to the external environment, to tackle CCSs.

The same specifics stand as reference elements for the development of the software learning environment described in the present work, which allowed to highlight both the soundness of the f-HIN model, as well as the feasibility and variety of learning questions the users may deal with using it. The improvement of the fHINscene application stems in fact from the general HIN theoretical framework, further refined according to the features of the f-HIN model, to provide a reliable computer-based platform to deliver education to (present and future) health professionals [21].

From an educational perspective, the use of digital technologies in student learning accounts for pedagogical innovation in health sciences education. It also addresses the growing need for a more systematic and organized approach to acquire, process, and assimilate information. Sophisticated skill sets, which are also becoming increasingly technology-oriented, are being required especially for the management of multimorbid patients [37,71,72]. In particular, the deployment of fHINscene to define, validate, and compare f-HINe networks can be ascribed, according to [14], to the category of Data Integration technologies, whose main concern is to identify clear clinical practice patterns in the accumulated clinical data about the treatment of many multimorbid patients. Moreover, they are adaptable to the possible changes in treatment standards, and the consequent therapeutic characteristics of the medical settings. More in general, the implementation of the HIN approach falls within the more general process of refinement of the case-based educational research methods (CBL) coming with it [73–76], as the learner can: (i) browse a clinical case over time; (ii) train to detect the evolutions and the interactions of the health issues; (iii) represent a clinical case of multimorbidity synthetically, with a diagram; (iv) develop the ability of CRM. Finally, since the HIN model is based on Petri Nets, it is possible to support an automatic assessment of the learner's performance in the execution of the clinical exercises [77].

Besides, an improvement was spotted for inquiry-based approaches as well, such as problem-based learning (PBL), as they revealed capable to prompt students to actively engage in knowledge construction and develop competencies across multiple contexts [78,79].

## 7. Conclusions and Prospects

In this paper, the f-HINe model has been presented as an effective way to draw clinical stories extracted from EHRs. Furthermore, the related fHINscene software has been described, which pursues the manifold objective of: (i) allowing the design of exercises for medical teachers according to the Case-Based Learning (CBL) approach; and (ii) providing learners with a system to learn clinical reasoning on real cases based on the evolution of a patient's health status over time, and, thus, the interaction between different clinical problems.

The f-HINe model has been tested several times in real use cases and with different users, as required by the Living Lab approach. Currently, f-HINe is used in some courses in one of the medical curricula at "Sapienza" University in Rome.

The next steps are:

- Testing of the f-HINe model and the related fHINscene software as an innovative teaching method in the Departments of Public Health, and Veterinary Medicine and Animal Production, at the "Federico II" University of Naples.
- Further investigations to improve the retrieval of clinical cases from the GP's EHRs that do not conform to the POMRs to make exercises based on the f-HINe diagram representing a real clinical case.
- Testing the possibility of incorporating the clinical history of a patient, drawn by f-HINe, with the related care pathway, to highlight the links between clinical-diagnostic

reasoning and the various diagnostic–therapeutic phases followed by the patient in a primary care environment based on the social-health integration.

- Validating the f-HINe model and the related fHINscene software as an environment for the assessment of the CRM ability, through exercises based on the interpretation, completion, and design of HINs. In this regard, the ability of the software to compare two HINs and measure their "distance" is fundamental, because of the high number of students attending clinical courses. The functionality of comparison between the teacher's and student's solution could greatly alleviate the teachers' workload for correcting the assignments. Studies are underway to extend the rules developed in the editor and comparator modules of the fHINscene software.

**Author Contributions:** Conceptualization, F.L.R. and F.C.; methodology, F.P., F.L.R., F.C., D.L., O.T.; software, F.P.; validation, F.L.R.; writing—original draft preparation, F.P., F.L.R., F.C., D.L., O.T.; writing—review and editing, F.P., F.L.R., F.C., D.L., O.T.; supervision, F.L.R.; funding acquisition, O.T. All authors have read and agreed to the published version of the manuscript.

**Funding:** This research was partially supported by the Board of Directors of the University of Naples Federico II through the FEDERICO project (Innovation in education for tenure–track researchers, grant EO/2020/933 of 11 June 2020) funded by the University of Naples Federico II.

**Acknowledgments:** Authors wish to thank the Italian Society of Medical Education; the students from "Sapienza" and "Federico II" Universities; the researchers and professors from "Federico II" University for the support provided in testing and validating both the HIN approach and the fHINscene software.

**Conflicts of Interest:** The authors declare no conflict of interest.

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
