# Peer review of "The Friendly Health Issue Network to Support Computer-Assisted Education for Clinical Reasoning in Multimorbidity Patients"

_electronics, doi:10.3390/electronics10172075_

Round 1
Reviewer 1 Report
SUMMARY BY THE REVIEWER AFTER READING THE MANUSCRIPT :
The authors put forth the argument that it is necessary to be able to anticipate the possible evolutions of a patient's overall health state and to recognize the interactions between the concurrent health issues and their treatments in order to practice clinical reasoning in multimorbidity conditions. They submit that this approach, which uses Petri Nets to represent and describe the network of health issues (HIs) that affect a person throughout their life and how they evolve over time, makes it easier to acquire such competencies. It represents graphically the network of health issues (HIs) that affect a person throughout their life and how they evolve over time. A more user-friendly version of HIN, known as f-HIN (friendly HIN), which is based on the same mathematical properties as HIN, has been developed to provide a more immediate and user-oriented interface to the algorithm. It has also been designed to be embedded in a software learning environment that allows for the creation of an f-HIN diagram, the verification of its format correctness, and the design of clinical exercises for learners, including the computer-assisted assessment of their performance on the exercises. The purpose of this paper is to introduce and discuss the f-HIN model, as well as its application in educational settings. This section also introduces the main features of the software learning environment on which it was built, emphasizing the importance of this environment in helping medical teachers design and represent the context of a learning outcome, as well as handling the complex history of a multimorbidity patient, which will be conveyed in Case-Based Learning (CBL) exercises.
Dear authors, thank you for your manuscript. I enjoyed reading it. Presented are some suggestions to improve it:
(1) Your title in its present form is too generic. Please consider modifying the title of the manuscript so that it is easier for potential readers to find your study. Please include the words "Petri Nets–based HIN (Health Issue Network)" or "f-HIN (friendly HIN)" if it is possible.
(2) Petri Nets is a type of state machine. Please include a section to help the readers understand why Petri Nets–based HIN (Health Issue Network) was considered by the authors to be the most suitable type of HIN for the purpose of your study to analyze health issues (HIs), compared to other types of state machines to model HINs? To help the readers, please kindly consider including a section to mention other types of state machines that may also be used to model HINs, and then please explain why Petri Nets was ultimately selected by the authors for this study.
(3) Please include a section to discuss what were the challenges faced when using Petri Nets to model HINs, and how your team overcame those challenges. This would be very beneficial to the readers as they would be able to learn from your expert knowledge.
(4) To improve the impact and readership of your manuscript, the authors need to clearly articulate in the Abstract and in the Introduction sections about the uniqueness or novelty of this study, and why or how it is different from other similar studies.
(5) Please substantially expand the discussion section, and compare your results to the ones found in similar studies. In particular, please cite more of the journal papers published by MDPI.
(6) Many of the references cited are not yet properly formatted. Currently, the DOIs are all missing from the citations of journal papers in the References section. For the references, instead of formatting "by-hand", please kindly consider using the free Zotero software (https://www.zotero.org/), and select "Multidisciplinary Digital Publishing Institute" as the citation format, since there are currently 57 citations in your manuscript, and there may probably be more once you have revised the manuscript.
Thank you.
Reviewer 2 Report
Dear author/s
I read with great interest your manuscript, which seems to be quite organized, however, the manuscript needs significant/major or minor changes that can be significantly improved. Below you will find some major or minor points in the manuscript which needs clarification, refinement, reanalysis, rewrites or/and additional information and suggestions for what could be done to improve it.
Reading your manuscript has given me the feeling that the sections are not connected and there is no continuous flow. Also, your manuscript deals with computer–assisted education as a concept, but this seems to be absent or unclear. Kindly include it.
In addition, the manuscript lacks a proper introduction that provides the sufficient context and adequate overview of the background. To help you, I quote some questions (as list of points) so that it can be included in your introduction:
-What is the importance of making this manuscript/contribution that it brings to the literature in the field?
-Why should readers be interested?
-What problem/ gap resolve/fill this manuscript?
-To fill this gap (resolve this problem) what solution/intervention/benefits does this manuscript bring? (in other words, how the proposed study will remedy this deficiency/gap/problem and provide a unique contribution to the literature).
-What is the research question which address to the purpose of the manuscript?
Based on the form of the manuscript, its type is considered a type of "Reviews". This journal suggests that you follow the PRISMA guidelines when reviewing, however you can follow any other way, where you have to justify it through reference. Kindly take a look at other review manuscripts in this journal to get an idea.
I would suggest you if you agree to create a section that will include the theoretical background of this work (as section 2), which will summarize the review of the literature, the concepts that you are negotiating and so on. In this section you can include various sections that describe the content of your manuscript. Also, this section may include sections that you already have (e.g., section 2 - The HIN approach).
New articles/researches have been published in the last 3 years, so it would be better to review the bibliography or add a more recent bibliography.
Kindly re-read your manuscript again with a clear mind and make the necessary corrections.
I would advise you not forget to read the instructions for the authors very carefully.
After the revision of your manuscript you may need to revise the other sections aw well as revise or re-edit the abstract and/or even the title. Kindly check for grammatical errors.
As a final comment, I recommend an in-depth revision of your manuscript.
Round 2
Reviewer 2 Report
Dear author/s,
I re-read your manuscript with great interest again.
Congratulations for the effort made to improve the work.
However, there are still some issues that I noticed that could be improved.
Kindly re-read your manuscript again with a clear mind and make the necessary corrections.
Based on the format of your manuscript, it is considered a type of "review" and not "article" (based on the authors' instructions). Please change the type of manuscript from "article" to "review" (line 1).
Subsection 2.2 looks a bit poor compared to the rest. I would recommend you if you could develop it a little more.
Please check the acronyms throughout your manuscript, because in some places they are not presented correctly.
As a final minor recommendation, kindly re-check for grammatical errors, and new publications that could form part of the manuscript.
Author Response
Please see the attachment.
Many thanks for helping us to improve the manuscript
